# Future Pandemic Influenza Virus Detection Relies on the Existing Influenza Surveillance Systems: A Perspective from Australia and New Zealand

**DOI:** 10.3390/tropicalmed4040121

**Published:** 2019-09-23

**Authors:** Lance C. Jennings, Ian G. Barr

**Affiliations:** 1Pathology and Biomedical Sciences Department, University of Otago, Christchurch 8011, New Zealand; 2WHO Collaborating Centre for Reference and Research on Influenza, VIDRL, Peter Doherty Institute for Infection and Immunity, Melbourne, Victoria 3000, Australia; Ian.Barr@influenzacentre.org; 3Department of Microbiology and Immunology, Peter Doherty Institute for Infection and Immunity, University of Melbourne, Victoria 3000, Australia

**Keywords:** influenza, surveillance, seasonal, pandemic, preparedness

## Abstract

The anniversary of the 1918–1919 influenza pandemic has allowed a refocusing on the global burden of influenza and the importance of co-ordinated international surveillance for both seasonal influenza and the identification of control strategies for future pandemics. Since the introduction of the International Health Regulations (IHR), progress had been slow, until the emergence of the novel influenza A(H1N1)2009 virus and its global spread, which has led to the World Health Organization (WHO) developing a series of guidance documents on global influenza surveillance procedures, severity and risk assessments, and essential measurements for the determination of national pandemic responses. However, the greatest burden of disease from influenza occurs between pandemics during seasonal influenza outbreaks and epidemics. Both Australia and New Zealand utilise seasonal influenza surveillance to support national influenza awareness programs focused on seasonal influenza vaccination education and promotion. These programs also serve to promote the importance of pandemic preparedness.

## 1. Introduction

The anniversary of the 1918–1919 influenza pandemic has allowed for a refocusing on the global burden of influenza and the importance of co-ordinated international surveillance for both seasonal influenza and the identification of control strategies for future pandemics. Progress has been slow since the revision of the International Health Regulations (IHR) in 2005 [1], which came into force on 15 June 2007, and called for strengthened surveillance for all events that constitute a “public health emergency of international concern (PHEIC).” Pivotal to these regulations are the surveillance and response capabilities that countries must establish and maintain to allow the detection, assessment, and reporting of disease events both nationally and internationally to the WHO within 48 h. While any urgent event can be assessed for notification under the 2005 IHR, there is mandatory notification for four diseases with the ability to cause serious public health impacts and to spread internationally. These include: a single case of smallpox, poliomyelitis due to wild type poliovirus, human influenza caused by a new subtype, and severe acute respiratory syndrome (SARS), and any cases of these diseases must be immediately notified to WHO, irrespective of the context in which they occur.

The emergence of the novel influenza A(H1N1)pdm2009 strain in April 2009 and its subsequent rapid global spread has highlighted the need to better understand the epidemiology of influenza viruses, especially the disease incidence and severity at the start of a novel event. The WHO has subsequently developed guidance documents for Global Epidemiological Surveillance Standards for Influenza [2] and global surveillance during an influenza pandemic [3]. The latter was updated in 2017 [4] and guidance on determining the severity during a pandemic, i.e. PISA (or Pandemic Influenza Severity Assessment tool) [5] is listed in Figure 1. These recent documents provide clear definitions of the new WHO pandemic phases and give guidance to countries on severity and risk assessments, surveillance procedures, and essential measures for the determination of national pandemic responses.

However, the greatest burden of disease from influenza actually occurs between pandemics, during seasonal outbreaks and epidemics, with annual global influenza-associated respiratory deaths recently being estimated to affect between 291,243–645,832 individuals (4·0–8·8 per 100,000) annually [8]. By contrast, the 2009 H1N1 pandemic was estimated to account for some 201,200 respiratory deaths (range 105,700–395,600) [9]. This current paper outlines influenza surveillance activities in New Zealand and Australia, countries that recognise the importance of using seasonal influenza surveillance and annual control strategies for both national and community influenza awareness, education, and pandemic preparedness.

## 2. Purpose of Influenza Surveillance

The purpose of influenza surveillance is to minimize the impact of the disease by providing useful and timely epidemiological information to public health authorities to allow the initiation in near real time intervention and preventative measures for the control of influenza. Surveillance activities can range from the simple, such as the laboratory testing of clinical respiratory samples for influenza viruses, through to the complex, with one or more of the following systems being used: regional or national sentinel surveillance programs that detect influenza-like illness (ILI) and supplies respiratory samples for laboratory testing or, samples being tested on site using point-of-care tests in the emergency departments of hospitals or in outpatient settings. Complex systems may include hospital admissions due to influenza and ultimately deaths related to or caused by influenza. The level and complexity of the surveillance activities for influenza is often dependent on the resources available to that city, state, or country. Ongoing surveillance allows the establishment of historical seasonal activity trends and a range of expected values for the comparison of outbreaks related to circulating viruses or to new viruses, to allow assessment of disease incidence and severity. At the national level, especially in temperate climates where the seasonality of influenza is well-defined, ongoing laboratory surveillance data can assist seasonal influenza awareness programs, associated influenza immunisation in at-risk groups, and antiviral drug stocking. At the global level, surveillance is needed to provide candidate viruses for influenza vaccine production and further antigenic and genetic characterisation is also needed to investigate unusual outbreaks or cases that may involve detection of potential pandemic viruses. In addition, laboratory surveillance is also used to determine the levels of antiviral drug resistance that are prevailing in circulating influenza viruses and potential pandemic viruses.

## 3. Global Influenza Surveillance

The World Health Organization (WHO) Global Influenza Surveillance and Response System (GISRS) is a worldwide network of laboratories that was established in 1952 to monitor changes in influenza viruses circulating in the human population, with the aim of reducing its impact through the use of vaccines, and more recently, antiviral medications [10]. The WHO Collaborating Centre for Reference and Research on Influenza in Melbourne is one of five such Collaborating Centres (CCs), with the others being located in Atlanta, Beijing, London, and Tokyo. Below these WHOCCs is a network of National Influenza Centres (NICs) (*n* = 143) in 113 countries [11], which are supported by their respective governments, and these laboratories are also supplemented with samples provided through large public and private diagnostic laboratories. One of the main purposes of maintaining this network is to have comprehensive monitoring of the antigenic and genetic changes in circulating human influenza viruses, and to make bi-annual recommendations on which influenza strains should be included in the influenza vaccine for the upcoming winter season in either the northern or southern hemisphere. Outputs from this network can be seen online on the WHO website with WHO FluNet (https://www.who.int/influenza/gisrs_laboratory/flunet/en/), which graphs the number of laboratory cases for the various types of influenza over time by country, region, hemisphere, or globally, and the WHO Flu ID website (https://www.who.int/influenza/surveillance_monitoring/fluid/en/), which is a global platform for data sharing that links and displays regional influenza epidemiological data. Both Australia and New Zealand contribute to FluNet (Figure 2).

## 4. Surveillance in Australia and New Zealand

Within Australia there are three NICs (PathWest, Perth; ICPMR, Sydney; and VIDRL, Melbourne) while in New Zealand there are two NICs (ESR, Wellington and Auckland Hospital, Auckland). These NICs and a number of other public health, hospital, and private laboratories support the Melbourne CC by providing influenza-positive samples and isolates each year along with many other laboratories in the southeast Asian and Oceania regions. The success of this regional network can be seen in the high number of Candidate Vaccine Viruses (or CVVs) for both egg-derived and cell-derived influenza vaccine viruses that have been made available and utilised over the past two decades from this region [12]. As well as providing influenza samples, a number of the supplying laboratories perform varying levels of analysis of the viruses ranging from simple typing (influenza A or B) to subtyping of the influenza A viruses (H1N1, H3N2, H5Nx, H7N9) usually by real time RT-PCR testing, or the determination of influenza B-lineages, and some by sequencing of the influenza genes or performing antiviral resistance analysis by either sequencing or in vitro assays.

The NICs and other laboratories in Australia and New Zealand also contribute to comprehensive national influenza surveillance programs. In Australia, all laboratory-confirmed influenza cases are reported to the NNDSS (Australian National Notifiable Diseases Surveillance System) database [13], which collects information on location (postcode) sample date, age, sex, influenza type, and subtype. This data is supplemented by the nationwide general practitioner (GP) influenza surveillance network ASPREN (Australian Sentinel Practices Research Network) [14], which records influenza-like illness (ILI) in the community and sends a selection of respiratory samples taken from patients with ILI attending GP surgeries to be tested for influenza. Another network, FluCAN (Australian Influenza Complications Alert Network) [15], records hospital attendance and admissions from patients suffering from influenza across all age groups. These networks allow the estimation of influenza vaccine effectiveness (VE) against medical attendance for community-based individuals [16] and against hospital admission each year [17]. These detection methods are supplemented by a national online weekly reporting surveillance system for cough and fever (Flutracking) [18], as well as a national medical hotline [19], which records enquiries related to ILI. There is also national death data due to influenza, as well as excess mortalities due to influenza and pneumonia being determined for the most populous state of Australia, New South Wales (NSW). These data are collated into the Australian Influenza Surveillance Report by the Australian Department of Health, which publishes fortnightly reports on the web from May to November each year [20]. Would these measures be sufficient to detect or track the introduction or outcomes of a new influenza pandemic as it spread throughout the country? Probably not, as detection of a new influenza pandemic virus would be dependent on general practitioners’ willingness to test samples from patients with ILI. If the cases were mild, then this is less likely to happen, as many general practitioners are reluctant to have laboratory testing performed on their patients with ILI and they rely on their knowledge and clinical presentation to make a diagnosis. If the case was more serious, or even fatal, then the chances of the patient being tested in a hospital setting and diagnosed correctly are increased, but a novel virus might still be mistyped in the laboratory, unless they had specific reagents. In addition, there are the inherent delays of at least 2–3 weeks in most of these systems, and they also lack the ability to provide sufficient data on sub-populations of most interest, such as pregnant women, indigenous populations, obese individuals, and individuals with other predisposing conditions who might have higher morbidity and mortality in a pandemic situation. Hence, the Australian Department of Health has explored the use of “first few hundred” studies and are also trialing the WHO PISA process as a potential addition to the current monitoring of seasonal influenza in case of a new influenza pandemic or a human outbreak involving a novel virus, so that they can be better informed. The main document for the management of influenza pandemics in Australia was revised in September 2014 and is known as the Australian Health Management Plan for Pandemic Influenza (AHMPPI) [21] This plan outlines Australia’s strategy to manage an influenza pandemic and minimise its impact on the health of Australians and our health system.

In New Zealand, the national influenza surveillance program includes acute respiratory illness surveillance in selected clinical settings to extend the knowledge base of the burden of respiratory disease and provide a robust system for pandemic preparedness. Sentinel surveillance involves 80–90 general practices across the country (1 practice per 50,000 population), the recruitment of patients presenting with an ILI, electronic collection of ILI data, and samples collected for respiratory virus testing from cases of ILI. In addition, severe acute respiratory infection (SARI) surveillance is carried out in four hospitals in Auckland, where respiratory samples are also routinely taken for testing for influenza and other respiratory pathogens. This data is published online weekly by ESR as a series of dashboards in the New Zealand Influenza Intelligence Report [22]. These systems allow influenza VE estimations to be made for general practice attendance and hospital admission each year. There is also the National Healthline, which uses electronic clinical decision software to record ILI, and the national recording of excess mortality for pneumonia and influenza. Laboratory-based virologic surveillance for influenza and other respiratory viruses continues year-round and is reported weekly [23]. The question is whether these measures would be sufficient in the event of the emergence of a novel pandemic virus? In 2009, New Zealand was the first country in the southern hemisphere to report the importation of the influenza A(H1N1) pdm 2009 virus, following the return of a group of high school students from Mexico on 25 April 2009. The New Zealand Influenza Pandemic Action Plan (NZIPAP) [24] provides an escalation pathway for enhanced surveillance, utilising hospital emergency departments and sentinel general practices once there is a pandemic alert. However, in 2009, it was the clinical astuteness of community general practitioners that was responsible for the identification of these first cases and the triggering of the NZIPAP and public health response [25,26].

## 5. Surveillance and Pandemic Preparedness

Of global concern is the emergence of a new novel influenza virus with pandemic potential. Surveillance programs for influenza alone may not accomplish the early detection and recognition of the initial event, however, they have considerable value in establishing the infrastructure necessary to respond to a pandemic both at national and community levels. Australia and New Zealand have national influenza awareness programs focused on seasonal influenza vaccination education and promotion. However, these programs also serve to promote the importance of pandemic preparedness. One example of this integrated approach is currently operating during the influenza season (winter) in Canterbury, New Zealand, where influenza and other respiratory virus laboratory surveillance data, along with Christchurch Hospital Emergency Department attendance and ICU admission data, is electronically circulated weekly to every general practice. This is a strategy used by the Canterbury Primary Response Group (CPRG) to communicate key messages to the community healthcare workforce about respiratory virus activity, preparation for influenza vaccine administration, and how to respond to evolving seasonal influenza activity. Supporting this initiative is a primary care response plan, which includes defined influenza season triggers for primary response escalation, the streaming of ILI patients to pre-designated clinics, advice for community nursing, aged-care and long-term care facilities, telephone triaging, and ambulance services. The primary objective of this plan is to optimise the use of community health care resources to protect secondary care services (Figure 3). Despite the considerable impact of the H1N1 pdm 2009 virus in Canterbury, health care services were not overwhelmed in 2009, the key lessons being the importance of intelligence and communications (the effective use of surveillance data), and preparing and working together across the sector [27]. Other countries, like China, have taken a more pro-active approach by setting up specialised clinics for cases of pneumonia with an unknown cause, in the hope of early detection of severely ill patients that may have acquired a novel influenza virus infection, such as the A(H7N9) virus that emerged in 2013 and has infected over 1500 people and killed over 600 of these people to date [28]. These systems remain in place, despite the virtual disappearance of human cases of A(H7N9) since mid-2018 after an extensive poultry vaccination program, with a bivalent H5/H7 vaccine beginning in China in September 2017 [29].

## 6. Conclusions

Seasonal influenza and other respiratory surveillance can provide information for understanding the contribution of these respiratory viruses to the global burden of disease, and the value of future potential virus-specific treatments. They also provide platforms for strengthening public health infrastructure, as has been seen in the western Pacific region in the past few decades [30]. The utilisation of routine surveillance to strengthen community influenza awareness initiatives is pivotal for successful pandemic preparedness and during a pandemic as it moves to each and every individual country across the globe.

## Figures and Tables

**Figure 1 tropicalmed-04-00121-f001:**
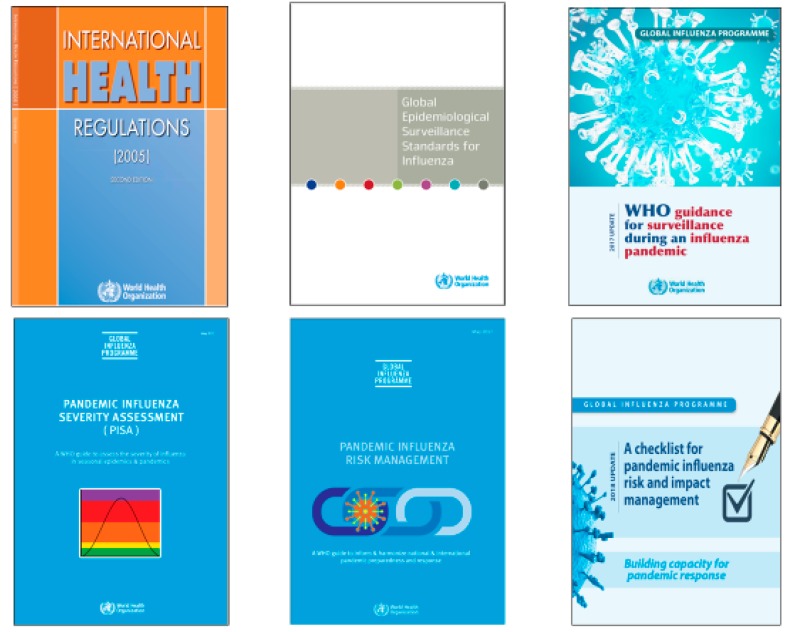
WHO documents which provide clear definitions of the new WHO pandemic phases and give guidance to countries on severity and risk assessment, surveillance procedures, and the essential measures for the determination of national pandemic processes [1,2,4,5,6,7].

**Figure 2 tropicalmed-04-00121-f002:**
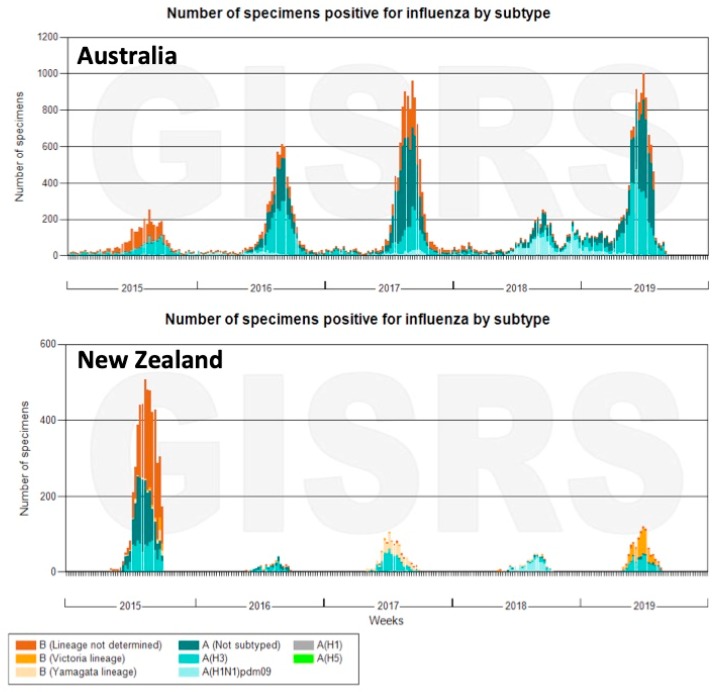
Laboratory-confirmed cases of influenza by type, subtype, and B-lineage reported to WHO FluNet by Australia and New Zealand National Influenza Centres from 2015–2019 (up to 4 September 2019) see https://www.who.int/influenza/gisrs_laboratory/flunet/charts/en/.

**Figure 3 tropicalmed-04-00121-f003:**
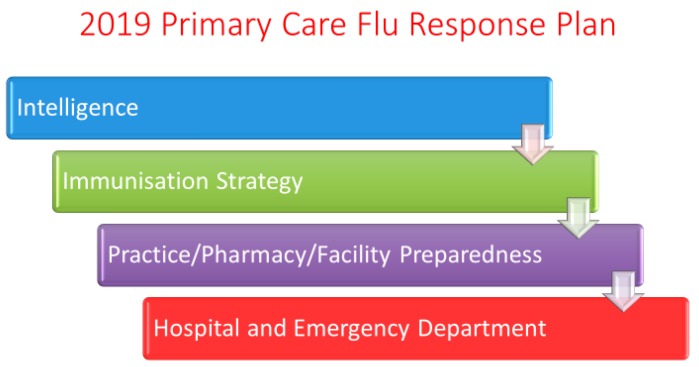
The Canterbury Primary Response Group (CPRG) Primary Care Response Plan aims to optimise the use of community health care resources to protect secondary care services. Pivotal to this strategy is pre-seasonal planning, the use and communication of intelligence (surveillance) data, and working together across the health sector.

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
