# Peer review of "Future Pandemic Influenza Virus Detection Relies on the Existing Influenza Surveillance Systems: A Perspective from Australia and New Zealand"

_tropicalmed, 2019, doi:10.3390/tropicalmed4040121_

Round 1

Reviewer 1 Report

Abstract – too wordy and the last sentence is incomplete. I recommend reducing the number of prepositional phrases to make the abstract more readable.

In general, many sentences are very long.

Section 2 – It would be useful to explain how general surveillance and laboratory surveillance are actually done. What clinical observations or information are collected? What specimens for laboratory analysis are collected? (This can be described n very brief terms.)

Section 3 – brief examples?

Section 4 – Very good descriptions of specific programs

The paper is interesting and well-written.

Author Response

Reviewer #1. Comments and Suggestions for Authors

Abstract – too wordy and the last sentence is incomplete. I recommend reducing the number of prepositional phrases to make the abstract more readable.

Response: The last incomplete sentence has been removed.

In general, many sentences are very long.

Response: Recommendations of Reviewer #2 have essentially been followed and sentences not shortened.

Section 2 – It would be useful to explain how general surveillance and laboratory surveillance are actually done. What clinical observations or information are collected? What specimens for laboratory analysis are collected? (This can be described n very brief terms.)

Response.  We have added a brief explanation of the types of surveillance that can be performed and added that the testing is done on respiratory samples.  To go into this in more detail is beyond the scope of this short review article.

Section 3 – brief examples?

Response: The global influenza surveillance network has been outlined and referenced, however some of the outputs from GISRS have been included to address this suggestion.

Section 4 – Very good descriptions of specific programs

The paper is interesting and well-written.

Reviewer 2 Report

This isn a very safe paper that will be a good reference article for the readership. There are a few points where improved clarity is required, one nonsensical sentence in the Abstract, and the Conclusion over-reaches in places (given what is actually covered in the article body). An annotated pdf explains.

Author Response

Reviewer #2. Comments  and Suggestions for Authors

a).The last sentence of the abstract does not make English sense.

Response: This sentence has been removed.

b) Please add (PHEIC) as many readers will recognise the abbreviation.

Response: Inserted as requested.

c) please add a clearer emphasis on the timeliness of the data so that interventions are also initiated on-time, as opposed to after the event or after the point of maximal impact.

Response: the wording has been modified to achieve this as recommended.

d) I think you are referring to data on vaccine uptake here? Please clarify because general data on influenza activity arrive 6-12 months after vaccine for the same season has been ordered. So the data are not timely for this purpose, whereas they can be for monitoring and  driving uptake in risk groups.

Response: This has been clarified in the manuscript. Both Australia and New Zealand have very active influenza awareness programs linked to vaccine roll-out. These programs rely heavily on laboratory surveillance data for messaging related to vaccine uptake.

e) general editorial point> All of these data are.... Data are plural.

Response: corrected as requested.

f) quantify typical delays please?

Response: Time period added,

g) Please elaborate here that there has to be willingness of physicians to test, and many still do not, even for SARI.

Response:  This has been added

h) routinely?

Response: inserted

i) It would be right and proper to ask the authors to build in further remarks that H7N9 has died right back since mid-2018 in China after the poultry vaccine campaign began. At present a reader could imagine or assume these cases are ongoing with vigour whereas they are not. But granted I don't think we'd want to convey any sense that we are comfortable the threat has gone away.

Response: sentence inserted o address this.

j) Which "community"? do you mean Western Pacific. Or do you mean community as opposed to hospital settings? If the latter, how do you know this? And how is it relevant to the Conclusions?

Response: This sentence has been deleted (see below (k))

k) This final sentence clearer and true. But how is it a conclusion for what you have been discussing in the article. I strongly suggest  stop after the sentence mentioning Western Pacific Region. The keep "the utilisation..." as your last sentence. At present you start a whole new thread (in two sentences) that does not closely relate to surveillance concepts you have discussed in the main body of the paper.

Response: Suggestion followed and sentence deleted.

Reviewer 3 Report

This is an interesting overview of influenza surveillance in Australia and New Zealand. The authors make an important point that seasonal influenza surveillance is an important process for detecting a pandemic, but is probably not optimized to do so. 

However, its unclear what new information this paper provides. In addition, the paper needs extensive editing for grammar (punctuation, capitalization, sentence structure). 

Perhaps a more systematic evaluation of current surveillance activities, with clear documentation of gaps and potential solutions, would be a way to improve the paper. 

Author Response

Reviewer #3. Comments and Suggestions for Authors

This is an interesting overview of influenza surveillance in Australia and New Zealand. The authors make an important point that seasonal influenza surveillance is an important process for detecting a pandemic, but is probably not optimized to do so. 

However, its unclear what new information this paper provides.

Response:It is the intention of this manuscript to highlight the importance of routine influenza surveillance and its role in pandemic preparedness. Australian and New Zealand examples are included to highlight the value that these countries place on routine surveillance and utilise the data generated.

In addition, the paper needs extensive editing for grammar (punctuation, capitalization, sentence structure). 

Response: The suggestions of Reviewer #2 have been followed for this purpose.

Perhaps a more systematic evaluation of current surveillance activities, with clear documentation of gaps and potential solutions, would be a way to improve the paper.

Response: this was not the purpose of this manuscript.

Round 2

Reviewer 3 Report

Although minor changes have been made, the major issues with this paper remain. It may be useful for readers to have a picture of influenza surveillance in Australia and New Zealand, however the abstract doesn't say anything about this since the last sentence has been deleted. 

I don't think Figure 1 is necessary and I don't see that Figure 2 is referenced. 

Author Response

Please see the attachments X 2:

Reviewer 2 Round 3: Response to Reviewer

Tropicalmed -535364 (1) : Amended manuscript
